# *Ago1* Affects the Virulence of the Fungal Plant Pathogen *Zymoseptoria tritici*

**DOI:** 10.3390/genes12071011

**Published:** 2021-06-30

**Authors:** Michael Habig, Klaas Schotanus, Kim Hufnagel, Petra Happel, Eva H. Stukenbrock

**Affiliations:** 1Christian-Albrechts University of Kiel, Environmental Genomics, Am Botanischen Garten 1-11, 24118 Kiel, Germany; mhabig@bot.uni-kiel.de (M.H.); klaas.schotanus@gmail.com (K.S.); schreibmichkim@web.de (K.H.); 2Max Planck Institute for Evolutionary Biology, August-Thienemann-Str. 2, 24306 Plön, Germany; 3Max Planck Institute for Terrestrial Microbiology, Karl-von-Frisch Strasse 10, 35043 Marburg, Germany; happelp@mpi-marburg.mpg.de

**Keywords:** RNA interference, pathogenicity, chromosome biology

## Abstract

In host-pathogen interactions RNA interference (RNAi) has emerged as a pivotal mechanism to modify both, the immune responses of the host as well as the pathogenicity and virulence of the pathogen. In addition, in some fungi RNAi is also known to affect chromosome biology via its effect on chromatin conformation. Previous studies reported no effect of the RNAi machinery on the virulence of the fungal plant pathogen *Zymoseptoria tritici* however the role of RNAi is still poorly understood in this species. Herein, we elucidate whether the RNAi machinery is conserved within the genus *Zymoseptoria*. Moreover, we conduct functional analyses of Argonaute and Dicer-like proteins and test if the RNAi machinery affects chromosome stability. We show that the RNAi machinery is conserved among closely related *Zymoseptoria* species while an exceptional pattern of allelic diversity was possibly caused by introgression. The deletion of *Ago1* reduced the ability of the fungus to produce asexual propagules *in planta* in a quantitative matter. Chromosome stability of the accessory chromosome of *Z. tritici* was not prominently affected by the RNAi machinery. These results indicate, in contrast to previous finding, a role of the RNAi pathway during host infection, but not in the stability of accessory chromosomes in *Z. tritici*.

## 1. Introduction

RNA interference (RNAi) involves widely conserved pathways present in many eukaryotes in which small RNAs (sRNAs) modify gene expression transcriptionally or post-transcriptionally [1,2,3]. The canonical RNAi machinery contains a double-stranded RNA endonuclease (Dicer or Dicer-like) that can also possess RNA helicase activity, RNA-binding protein(s) (Argonaute), and, in some species, an RNA-dependent RNA polymerase (RdRP) [4]. Small dsRNAs of 20–30 nt are generated by the Dicer RNase III domains from double stranded RNAs. These small dsRNAs are then bound by an Argonaute protein which is part of the RNA induced silencing complex (RISC). Here the passenger strand of the small RNA duplex is removed and the small single stranded sRNA is used by the RISC complex as a guide for the recognition and degradation of RNAs [4,5]. Several groups of sRNA have been categorized, e.g., microRNAs (miRNAs) which are derived from single-stranded RNA precursors with stem-loop structures and small interfering RNAs (siRNAs), derived from long double dsRNAs [6]. RNAi is involved in a very diverse set of functions, ranging from defense against viral or transposon invasion to the regulation of developmental processes, immune responses to pathogens, and epigenetic modifications, to name but a few [4,7,8,9,10]. Interestingly, the number of Argonaute proteins varies considerably among taxa [11,12] and there is considerable variation within the fungal kingdom [13,14] with some species having lost the ability to produce sRNAs [2,15], e.g., *Saccharomyces cerevisiae*, as well as the smut fungus *Ustilago maydis* which both lack functional components of the RNAi pathway [16,17].

An emergent body of evidence supports the importance of RNAi in plant-pathogen interaction. Firstly, RNAi has been demonstrated to play a role in the endogenous regulation of plant immune responses [18,19,20], e.g., the downregulation of the auxin pathway in *Arabidopsis thaliana* upon detection of the PAMP molecule flg22 [19] (reviewed in [21]). Secondly, RNAi was also found to play an important role in the regulation of virulence in some pathogen species [21]; e.g., the rice blast fungus *Magnaporthe oryzae* produces sRNAs in infection structures during the initial infection [22]. Lastly, in some pathosystems it has been documented that sRNAs can be transferred between fungal and plant cells to modify the expression of target genes involved in plant immunity [23,24]. An intriguing example is the grey mold fungus *Botrytis cinerea* that produces and translocates sRNAs into its host (*A. thaliana* and *Solanum lycopersicum*) to hijack the host RNAi machinery and suppress the transcription of genes involved in the host defenses [25,26,27]. In addition to a role in virulence, small RNAs can also be involved in other genetic and epigenetic processes, such as the maintenance of chromatin states and proper chromosome segregation (reviewed in [28]). Epigenetic marks, like histone modifications can, in some fungal species, be directly affected by small RNAs [4]. This has been demonstrated by the deletion of Argonaute and Dicer encoding genes in the yeast *Schizosaccharomyces pombe* which resulted in the loss of heterochromatin [8]. The RNAi-involving formation of heterochromatin is initiated by the recognition and degradation of nascent long non-coding RNA transcript (lncRNA) [29,30,31,32]. This recognition of lncRNAs results in the RNAi-dependent methylation of H3K9, possibly via the involvement of AGO1 [33,34]. While there is a connection between RNAi and chromatin organization via histone modifications in some fungal species, it is not a general mechanism. The high degree of variability in the presence/absence and function of the RNAi machinery between fungal species suggests that these genetic processes can be mediated and regulated by different mechanisms in different species [2,15,28,35], but experimental studies beyond a few selected model-organisms are scarce.

In the wheat pathogenic fungus *Zymoseptoria tritici* the role of RNAi in pathogenicity as well as chromosome biology is still unresolved. *Z. tritici* is a hemibiotrophic pathogen that infects wheat and can cause dramatic losses in yield [36,37,38]. Recently, in order to address whether small RNAs play a role in the pathogenicity of *Z. tritici*, Kettles and colleagues identified and deleted all components of the canonical RNAi machinery in *Z. tritici* [39]. Surprisingly, they did not observe any qualitative phenotypic difference of *Dcl*, *Ago1* and *Ago2* mutants during infection of the susceptible wheat cultivar Bobwhite [39]. In a second study, Ma and colleagues assessed the putative impact of fungal small RNAs on host gene expression or immune response, but did not observe any evidence for such effects of fungal sRNAs [40]. Similarly, the role of RNAi in the chromosome biology of *Z. tritici* has so far not been investigated. The genome of this fungus includes a set of 13 core chromosomes and a variable number of accessory chromosomes (i.e., eight in the reference isolate IPO323), which are non-essential and subject to chromosome loss during mitosis and a meiotic drive during sexual reproduction [41,42]. The functional role of the accessory chromosomes in *Z. tritici* is still unknown [43] but these are enriched with the histone modification H3K27me3 [44]. Indeed, we could recently show that the histone modifications H3K9me3 and H3K27me3 affect the transmission of accessory chromosomes during mitotic divisions in opposite directions [45,46]. Since histone modifications are known to be affected by RNAi in other fungal species and in *Z. tritici* affect chromosome stability we hypothesized that RNAi, via H3K9me3 or H3K27me3, may affect the transmission of accessory chromosomes in *Z. tritici*.

In this study we elucidate the role of RNAi in pathogenicity and the chromosome segregation of the Septoria-tritici-blotch-causing wheat pathogen *Z. tritici*. The main questions in our study were: (i) Is the RNAi machinery conserved within the genus *Zymoseptoria*, supporting a conserved functional relevance? (ii) Is there a quantitative impact of the individual components of the RNAi machinery on the phenotype *in planta* and in vitro? (iii) Does the RNAi machinery affect the transmission of accessory chromosomes of *Z. tritici* during mitotic cell division? We address these questions using phylogenetic analyses, phenotypic characterization of RNAi mutants in vitro and *in planta* and an experimental evolution approach.

## 2. Materials and Methods

### 2.1. Identification of Dicer-Like and Argonaute Genes

Using the protein sequences of previously characterized Dicer-like and Argonaute encoding genes in *Neurospora crassa* [47], *S. pombe* [48] and *C. elegans* [49], homologs of the RNAi genes in the reference genome of the *Z. tritici* isolate IPO323 [36] using Basic Local Alignment Search Tool (BLAST) analyses [50] and RNA seq-based annotation were identified [51]. Chromosomal coordinates of the identified Dicer-like and Argonaute genes of *Z. tritici* IPO323 are summarized in Appendix A. Protein domains were annotated using InterProScan.

### 2.2. Phylogenetic Analysis

A total of 36 publicly available genome assemblies for different *Z. tritici* isolates as well as isolates of the closely related sister species *Z. ardabiliae*, *Z. brevis* and *Z. passerinii*; the latter used as outgroup (see Appendix A), were used to analyze the homologs to the IPO323 *Dcl* and *Ago1-4* genes using BLAST [50]. Homolog sequences were aligned using MUSCLE (v3.8.425). A neighbor-joining phylogenetic tree was constructed with Geneious Tree Builder (V2021.1.1) with 1000 bootstrap replicates using *Z. passerinii* as an outgroup. *Ago4* had no homolog in *Z. passerinii*; hence, *Z. brevis* (isolate Zb18110) was used as an outgroup for *Ago4*. Phylogenetic networks were constructed using SplitsTree (v 4.17.0) [52].

### 2.3. Fungal Material

Experiments were performed with a derivate of the *Z. tritici* reference strain IPO323 [36] that had spontaneously lost accessory chromosome 18 [53]. Since all deletion and complementation clones were constructed in and compared to this IPO323 ΔChr18 derivate, we expect no effect on the interpretation of the results in this study. Cells from −80 °C glycerol stocks were used as initial inoculum on YMS (4 g yeast extract, 4 g malt extract, 4 g sucrose, and 20 g agar per 1 L H_2_O) agar plates. Cultures were grown for four to six days at 18 °C, transferred to liquid YMS medium and grown for three days at 18 °C while shaking at 200 rpm. Fungal cells were grown in the dark. Cells were harvested by centrifugation at 16,000× *g* for 2 min (for DNA extraction) or 3000× *g* for 10 min (for plant infection). 

### 2.4. Deletion of Argonaute and Dicer-Encoding Genes

Homologous recombination constructs were based on the binary vector D0893pNOVpGpda_SDHB_H267YtTrpC [54] with hygromycin (hph) or geneticin (G418) resistance cassettes for deletion and complementation, respectively. We generated constructs with ~1000 bp upstream and downstream flanking sequences of the respective genes for homologous replacement with the deletion cassettes.

All constructs were created in vitro by Gibson assembly or overlap PCR [55]. Fungal DNA was extracted from cells by glass-bead homogenization using a previously described phenol-chloroform method [56]. Genomic DNA was used for PCR and Southern blot analyses according to previously published protocols [57]. We confirmed the correctness of the transformation cassettes by restriction analysis and Sanger sequencing. All primers are listed in Appendix A. Integration of constructs into the *Z. tritici* genome was conducted by *Agrobacterium tumefaciens*-mediated transformation as previously described using the *A. tumefaciens* strain AGL1 [58]. Correct integrations of the constructs were assessed first by an initial PCR screen (Appendix A). The identified deletion candidates were subsequently tested by Southern blot analyses (Appendix A). 

### 2.5. In Planta Virulence Assays

For the *in planta* phenotypic assays, we germinated seeds of the wheat cultivar Obelisk on wet sterile Whatman paper for four days before potting using the soil Fruhstorfer Topferde (Hermann Meyer GmbH, Rellingen, Germany). Wheat seedlings were further grown for seven days before inoculation. *Z. tritici* was grown as described above, centrifuged and resuspended in H_2_O. The cell number was adjusted to 10^7^ cells/mL in H_2_O + 0.1% Tween 20, and the cell suspension was brushed onto approximately five cm on the abaxial and adaxial side of the second leaf of each seedling. Inoculated plants were placed in sealed bags containing water for 48 h to facilitate infection through stomata. Plants were grown under constant conditions with a day night cycle of 16 h light (~200 µmol/m^2^*s) and 8 h darkness in growth chambers at 20 °C. Plants were grown for 21 days post inoculation at 90% relative humidity (RH). At 21 dpi the infected leaves were cut and taped to sheets of paper and pressed for five days at 4 °C before being scanned at a resolution of 2400 dpi using a flatbed scanner (HP photosmart C4580, HP, Böblingen, Germany). Scanned images were analyzed using an automated image analysis in Image J (v1.53e) [59] adapted from [60]. The read-out pycnidia/cm^2^ leaf surface was used for all subsequent analyses (Appendix A).

### 2.6. Experimental Evolution Experiment

The experimental evolution experiments were conducted similarly as previously described [42]. Two independent clones for each deletion and complementation were included. The list of included clones and strains is given in Appendix A. In brief, a single colony grown for one week at 18 °C on YMS agar was resuspended in 9 mL liquid YMS media and 2.5 mL of each was distributed into three test tubes and inoculated at 18 °C and 200 rpm for 3–4 days before 90 µL (representing 4% of the population) were transferred to a new test tube containing 2.5 mL liquid YMS media and further inoculated. The transferred volume was chosen to contain a sufficiently large number of fungal cells (>10^5^ cells) to minimize drift and therefore maximize the likelihood that the resulting populations were least affected by random effects. No selective pressure was specifically applied during the time course of cell propagation. Each week an aliquot was added with 25% glycerol and stored at −80 °C for long-term storage. After four weeks the experimental evolution experiment was stopped; hence, a total of eight transfers was conducted for each of the three replicates for each of the two independent clones of the deletion and complementation strains as well as the wildtype (see Appendix A for an overview of strains and replicates). For the verification of the presence of accessory chromosomes dilutions of long-term storage aliquots were inoculated onto YMS plates and 32 to 48 individual colonies were tested as described before [42]. An accessory chromosome was considered to be present when two independent subtelomeric regions at the opposite end of the chromosome could be amplified with chromosome-specific primers by PCR. An accessory chromosome was considered to be lost, if both independent subtelomeric regions were not amplified. The analysis was restricted to accessory chromosomes 14, 15, 16, 20 and 21 which had shown the highest loss rates during previous in vitro experimental evolution experiments [42].

### 2.7. Statistical Analysis

Statistical analyses were conducted in R [61] using the suite R Studio (version 1.0.143). *In planta* data inspection showed a non-normal distribution for pycnidia density (pycnidia/cm^2^). Therefore, we performed an omnibus analysis of variance using rank-transformation of the data [62] using the model: pycnidia density ~ strain * experiment * operator where applicable. Post hoc pairwise comparisons to the wildtype were performed using Tukey’s HSD [63].

## 3. Results

First, we confirmed the presence of four distinct Argonaute homologs and one Dicer homolog in *Z. tritici*, as was previously reported [39]. We therefore searched the genome of the reference isolate IPO323 for homologs of known RNAi-related genes and could verify the reported four homologs of Argonaute, all containing the PIWI domain, and one homolog of Dicer (Appendix A). Interestingly for Ago1, Ago3 and Ago4 we identified more than one group of highly distinct alleles when comparing homologous sequence from different field isolates of *Z. tritici*. Using phylogenetic analyses, we found that some these allele groups cluster with homologs in the closely related sister species *Z. ardabiliae*, *Z. brevis* and *Z. pseudotritici* suggesting the occurrence of introgression of distinct alleles into *Z. tritici* (Figure 1, Appendix A). Introgression was previously found to confer similar unusual patterns of genetic variation along the *Z. tritici* genome [64]. The distribution of isolates having distinct allele groups does not correlate with geographic locations indicating that the hypothesized introgression events have occurred prior to the continental dispersal of *Z. tritici*. For example, for Ago1, one allele group contains mostly isolates from Germany (Zt05, Zt148, Zt151, Zt153, and Zt154) and Israel (ST9ISY); the second group represents samples that were collected in the same regions in Germany (e.g., Zt150) and in neighboring countries (Switzerland, e.g., 1E4, Netherlands; e.g., IPO323) (Appendix A). For *Dcl* and *Ago2* we found only one group of alleles consistent with the species tree of *Z. tritici* and a monophyletic origin of these two genes (Appendix A).

### 3.1. Deletion of Dcl and Ago Genes in Z. tritici Does Not Impact Basic Growth

To assess the functional importance of RNAi in *Z. tritici* we deleted the *Dcl* and three of the four *Ago* genes independently in IPO323 using *Agrobacterium tumefaciens* mediated transformation. We confirmed the correct integration of the transformation constructs using Southern blot analyses (Appendix A). To test the impact of RNAi on cell wall integrity and basic growth we exposed the mutants to an in vitro stress assay including temperature stress and chemical stress agents. This assay revealed no phenotypic difference of the mutants when compared to wildtype for the Dicer and Argonaute mutants (Appendix A) in line with previous reports [39,40]. Next, we conducted a comparison of growth rates to determine if the RNAi gene deletions would confer slower growth. To this end we determined and compared the maximum growth rate (µmax) and the carrying capacity (K) of the *Dcl*, *Ago1*, *Ago2* and *Ago3* deletion and complementation clones with the wildtype over the course of six days in liquid culture in 96 well plates (Appendix A). Neither the maximum growth rate nor the carrying capacity differed significantly between deletion and complementation strains compared to the wildtype. In summary, the deletion of *Dcl*, *Ago1*, *Ago2*, and *Ago3* appeared to have no qualitative or quantitative effect on the basic growth characteristics of *Z. tritici* in vitro.

### 3.2. Ago1 Affects the Pathogenicity of Z. tritici in Planta Quantitively

We next asked if the RNAi machinery affects the infection of *Z. tritici in planta* in a quantitative manner. In a previous study, fungal deletion strains of *Dcl*, *Ago1*, and *Ago2* were shown to be fully pathogenic on the wheat cultivar Bobwhite [39]. Using another susceptible wheat cultivar, we verified the absence of a phenotype between the wildtype strain and the deletion and complementation strains of *Dcl* and *Ago2* (Appendix A). Moreover, we observed no effect of *Ago3* deletion on the ability to infect and produce pycnidia in the wheat cultivar Obelisk (Appendix A). However, deletion of *Ago1* significantly reduced the ability of *Z. tritici* to produce asexual fructifications (pycnidia) during infections of the wheat cultivar Obelisk (Figure 2). Two independent clones in which *Ago1* gene was deleted (clone Zt108#42 and Zt108#59a) were both attenuated in their ability to produce pycnidia and showed a significantly lower density of pycnidia when compared to the wildtype in three independent experiments (Figure 2A, ANOVA, *p* = 1 × 10^−7^, *p* < 1 × 10^−7^, see Appendix A for results of individual experiments). Complementation of these deletions with the IPO323 allele of *Ago1* at the native locus (clone Zt201#8 and Zt201#20) restored the wildtype phenotype. Based on this observation, we conclude that *Ago1* specifically affects the ability of *Z. tritici* to infect and propagate *in planta*, at least in some wheat cultivars. This effect of *Ago1* appears to be quantitative and independent of *Dcl* as we did not observe any phenotypic effect by deletion of the *Dcl* gene. 

We mined previously published expression data to assess if the RNAi genes are expressed during host infection and if *Ago1* is upregulated. Indeed, previously published RNA-seq data of IPO323 obtained at 4, 11, 13 and 20 days post inoculation (dpi) [65] indicates a higher expression of *Ago1* compared to *Ago2*, *Ago3*, *Ago4* and *Dcl*. However, for all five genes, there are no significant changes in gene expression levels during the time course of host infection (Appendix A).

### 3.3. The RNAi Machinery Does Not Affect the Transmission of Accessory Chromosomes

We hypothesized that the RNAi machinery could play a role in the transmission of accessory chromosomes during mitotic cell divisions. To test this, we set up an in vitro evolution experiment. We transferred 4% of cell populations to new medium twice a week over the course of four weeks (Figure 3A). At the end of the experiment, we then tested the presence of accessory chromosomes by PCR amplification of two subtelomeric regions located at opposite end of each of five accessory chromosomes (chr14, chr15, chr16, chr20 and chr21). In the wildtype, 1.35% of all tested accessory chromosomes were lost after four weeks of experimental evolution (12 of 904 chromosomes were lost) (Figure 3B). For the ∆*Ago1* mutant we found slightly lower rates of accessory chromosome loss, however, the differences were not significant (FDR adjusted Fisher’s exact test *p* = 0.28). Overall, we found no difference in the rate of chromosome loss for either of the deletion strains *Ago1*, *Ago2*, *Ago3* and *Dcl* when compared to the wildtype (Figure 3).

## 4. Discussion

In this study, we confirm the presence of genes encoding for Dcl and Argonaute homologs in the genome in *Z. tritici*, as well as in genomes of closely related sister species within the *Zymoseptoria* genus. This conversation of a putatively functional RNAi machinery in *Z. tritici*, as well as its sister species supports a functional relevance. In contrast to earlier studies, herein we show that the deletion of *Ago1* quantitatively reduced the ability of *Z. tritici* to produce pycnidia, supporting a role of RNAi in the pathogenicity of *Z. tritici*. Finally, we do not see any indication that the RNAi machinery affects chromosome stability in *Z. tritici*.

Using phylogenetic analyses, we find that Dcl and Argonaute homologs are present in *Zymoseptoria* ssp., however, we identify a particular pattern of non-monophyletic *Z. tritici* alleles for *Ago1*, *Ago3* and *Ago4*. A non-monophyletic pattern could be explained by the acquisition of a foreign gene by horizontal gene transfer. However, based on the sequencing conservation of the regions encoding the Argonaute genes, the introduction of a new Argonaute gene by horizontal gene transfer is unlikely. If horizontal gene transfer had occurred, the transferred Argonaute gene would most likely be integrated into another part of the genome and would not replace an already existing Argonaute gene. Instead we speculate that introgression is the most likely scenario underlying the distinct phylogenetic clusters of *Z. tritici.* In *Z. tritici* introgressions have occurred frequently through hybridization with other *Zymoseptoria* relatives and has affected approximately 5% of the entire genome [64]. 

So far the functional relevance of distinct *AGO* alleles in *Z. tritici* is unknown, but they may allow interaction with distinct targets. In 2012, Bernhardt and coworkers described a high evolutionary rate of RNAi related genes in the mosquito *Aedes aegypti* [66]. All the genes of the exo-siRNA and the miRNA pathway in *Ae. aegypti*, showed signatures of positive selection. A similar finding was shown in three *Drosophila* species (*D. melanogaster*, *D. simulans* and *D. yakuba*) [67]. Like in *Ae. aegypti*, the genes involved in the exo-siRNA pathway were found to be under positive selection and among the top 3% of fastest evolving genes in *Drosophila*. This is in contrast to the genes involved in the miRNA pathway which show a much lower rate of evolutionary changes. It was hypothesized that the differences in the selection of the two pathways is due to the function of the exo-siRNA pathway. The authors hypothesized that this pathway is under selective pressure due to infection with flaviviruses. These viruses are insect specific. The role of viral infection in *Z. tritici* is not known. Several fungi, like *Saccharomyces cerevisiae* and *Ustilago maydis* have lost complete RNAi pathways due to infection from the Killer virus [68]. It is unlikely that infection with a Killer virus has led to divergence of the genes in the RNAi pathways of *Z. tritici.* However, we speculate that an infection with another virus could have led to the diversification of the Argonaute genes.

In the fungal kingdom, species vary in the number of Argonaute genes considerably, ranging from one in *Cryptococcus neoformans* to nine in *Phlebia brevispora* [2]. The underlying cause for this variability is unknown and could be due to functional diversification and/or redundancy. In *N. crassa* the two known Argonaute genes have distinct functions, with SMS-2 being involved in meiotic silencing and QDE-2 being involved in quelling [69]. For the majority of Argonaute genes in other fungal species the extent of functional diversification and redundancy is so far unknown. Herein, we here speculate that the four Argonaute genes in *Z. tritici* might reflect functional diversification, but can also not exclude functional redundancy.

Most importantly, we identify an effect of the *Ago1* gene specific for infection *in planta*. The ability to produce pycnidia was significantly reduced in a quantitative manner, implying that mutants were still virulent but exhibited lower fitness. Interestingly, the *Ago1* gene also showed the highest expression of all four *Z. tritici* Argonaute genes during plant infection, further supporting relevance during host infection. The observed phenotypic effect of the *Ago1* deletion however contrasts with previous studies that found no effects of *Ago1* deletion on virulence [39] and no effect on the cleavage of wheat transcripts [40]. Possibly, the very high sample size in combination with automated phenotyping enabled us to detect these quantitative effects of *Ago1* deletions. Alternatively, the different susceptible wheat cultivar used in our study (cv Obelisk) compared to the previous study (cv Bobwhite) could account for the differences in the observed effect of deleting *Ago1*. Interestingly, the effect on the pycnidia production of *Ago1* was independent of *Dcl* and apparent only during growth *in planta* but absent during growth in vitro. Dicer-independent production of sRNAs has been reported for several organisms [70,71,72,73], as well as in *Z. tritici* [39]. In a focused analysis three out of four sRNAs produced by *Z. tritici in planta* showed a dicer-independent upregulation during infection pointing to the existence of a Dicer-independent mechanism to produce sRNAs in *Z. tritici* [39]. In total, 66 fungal sRNA were upregulated during *in planta* infection of *Z. tritici* [40], all of which would represent interesting targets to further dissect the mechanisms of dicer-independent effects of *Ago1* on the infection of *Z. tritici in planta.*

Finally, we do not see any effect of RNAi on the transmission of the accessory chromosomes in *Z. tritici*. RNAi is known, in some fungal species, to be involved in the methylation of histone tails [10,74] and DNA methylation [75], and in addition possibly centromere length [76]. The transmission of the accessory chromosomes of *Z. tritici* is to a large extent determined by the histone modifications H3K9me3 and H3K27me3 [42,43,44]. Interestingly, this appears to be a system sensitive to environmental changes, as relatively small changes in temperature have pronounced effects on the frequency at which these chromosomes are lost [42]. The absence of an impact of the RNAi machinery on the transmission of the accessory chromosomes is therefore surprising. Possibly, in vitro RNAi in *Z. tritici* is less important, a fact that is supported by the absence of any detectable effects on the growth morphology or growth rate. It is possible that RNAi plays a role in the transmission of the accessory chromosomes *in planta*, wherein accessory chromosomes are also lost frequently [42]. 

## 5. Conclusions

In conclusion, we here present evidence for the functional relevance of the RNAi machinery on the *in planta* development of the fungus *Z. tritici*. Future studies should address the underlying molecular mechanisms of this effect as well as the relevance of distinct allelic clusters of Argonaute genes. Our study has laid the basis for future studies aiming to dissect the relationship between RNAi and histone-modifications, for which *Z. tritici* is could be an ideal model as it contains both a putatively functional RNAi machinery as well as a wide spectrum of histone and DNA modifying enzymes. 

## Figures and Tables

**Figure 1 genes-12-01011-f001:**
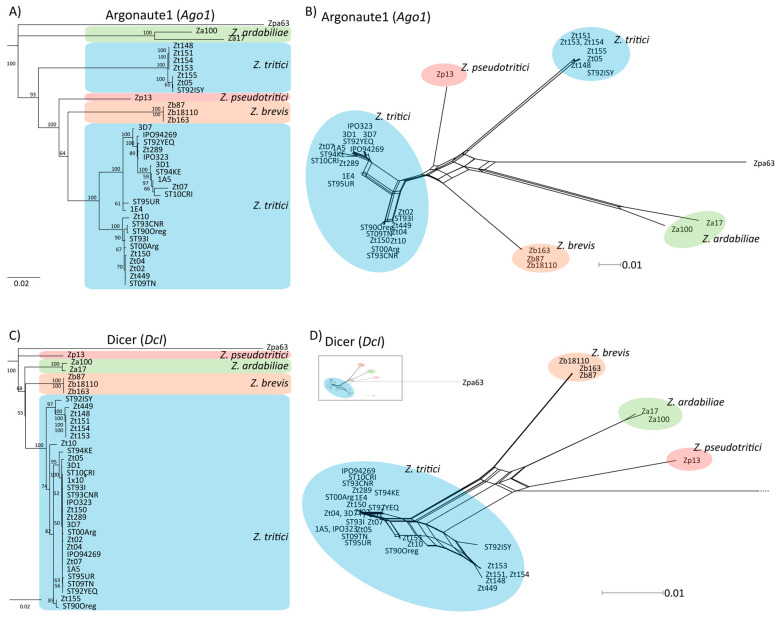
Phylogenetic relationship of *Ago1* and *Dcl*. (**A**,**C**) Neighbor-joining tree and (**B**,**D**) phylogenetic network of the respective homologs in *Z. tritici*, *Z. ardabiliae*, *Z. brevis*, and *Z. pseudotritici* and as an outgroup *Z. passerinii* (Zpa63). Support of nodes is indicated (% of 1000 bootstraps). (**C**,**D**). *Ago1* shows two distinct allele groups in *Z. tritici* (blue), with one allele group clustering with homologs in the sister-species of *Z. tritici* (*Z. ardabiliae* (green), *Z. pseudotritici* (red), and *Z. brevis* (orange)).

**Figure 2 genes-12-01011-f002:**
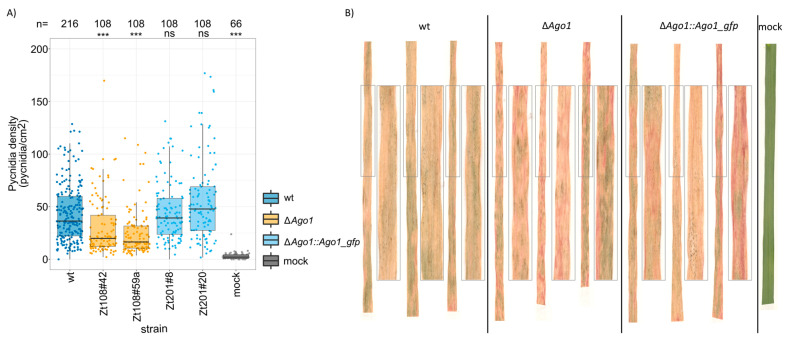
*AGO1* affected the ability of the fungus to infect and propagate *in planta*. Two independent *Ago1*-deletion strains (∆*Ago1*, orange) showed significantly lower density of pycnidia on the leaf surface at 21 dpi (days post inoculation) than the IPO323 wildtype (wt, dark blue). Complementation of the deletion by the wt *Ago1* allele in its genomic locus (light blue) restored the wt phenotype. Statistical significance was inferred by ANOVA on ranked Pycnidia densities using the model pycnidia density~strain * experiments * operator with a post hoc Tukey’s HSD in a pairwise comparison to the IPO323 wt. Pooled data from three independent experiments are shown. (**B**) Example pictures of infected wheat leaves with wt (IPO323), ∆*Ago1* deletion and ∆*Ago1::Ago1_gfp* complementation clones. Enlarged are representative portions of the leaf. Although all *Z. tritici* strains induced full necrosis of infected leaf portions the ∆*Ago1* deletion clones showed a reduced ability to produce pycnidia. (**A**) Categorized *p*-values of Tukey’s HSD post-hoc test on an ANOVA on ranked pycnidia density are shown (ns: not significant, ***: *p* < 0.0005).

**Figure 3 genes-12-01011-f003:**
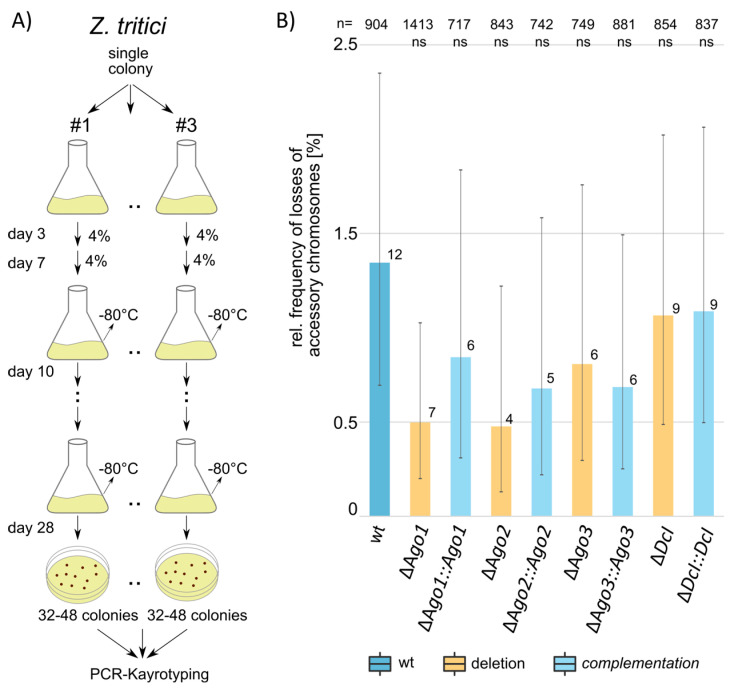
Mitotic transmission of accessory chromosomes appears to be not affected by the RNAi machinery. (**A**) Schematic depiction of the experimental evolution experiment. For each of the deletion and complementation strains two independent clones and one wildtype was used, each with 3 technical replicates (total of 41 replicates). (**B**) Relative frequencies of losses of accessory chromosomes for the wildtype and pooled for the two biological replicates of the indicated deletion and complementation strains. Categorized *p*-values of FDR adjusted *p*-values of a Fisher’s exact test are shown (ns: not significant). Error bars represent 95% Poisson confidence intervals.

## Data Availability

The data presented in this study are available in the Appendix A.

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
