# Peer review of "Ago1 Affects the Virulence of the Fungal Plant Pathogen Zymoseptoria tritici"

_genes, 2021, doi:10.3390/genes12071011_

Round 1
Reviewer 1 Report
In the manuscript “Ago1 affects virulence of the fungal plant pathogen Zymoseptoria tritici”, the authors show important work on the characterization of RNA interference system in this phytopathogen interaction with the host. The work follows a logical presentation. However, in my opinion, the authors need to further clarify the following points:
- The introduction is complete, however, it is too long and dispersed. The aims are well-identified yet the flow/relationships between them is missing. The results section needs also editing and there are some repeated phrases between them that, in my opinion, need to be rephrased.
- The materials and methods refer that a specific strain, missing chromosome 18 was used, yet there is no explanation for this fact. The authors need to explain how the number of cells was normalized between different cultivations. Dark? The authors should refer to the mock that is presented in figure 2. Please clarify in section 2.6 how many sub-cultures were done and if there is any pressure applied. Apparently, 8 sub-cultures were done, which seems little and may justify the results on this.
- Finally, in figure 2 refers to a mock (see point 2) which should be the inoculation of sterile equal to that used to inoculate Z. tritici, but without the fungal cells. Being so, it is surprising that there is a count of pycnidia in this assay even if it is just an artifact from the counting method. Otherwise, this suggests contamination with fungal cells from uncontrolled sources and makes the counting for pycnidia produced by the wt and mutant not so robust... Also, the images of the leaves are not intuitive and it is hard for me to see alterations in the pathogenicity between wt and mutants. The mutants despite having a smaller pycnidia count show the same phenotype? This seems to contradict the statement on line 333 (even if this is based on the pycnidia number). If the effect is not seen at that time point may be earlier effect is easier to spot.
Minor comments (
- i) sometimes the italics on the name of the fungus, In planta is missing;
- ii) superscript of the numbers and units are sometimes missing e.g. lines 177, 181, 187;
iii) Fig 3A- the workflow of the evolution can be supplementary information;
- iv) The gel presented in figure S3C, should if possible be repeated to exclude the lanes that are not related to the present work.
Reviewer 2 Report
The manuscript verifies the research hypotheses in a very thorough and very advanced manner.
Materials and Methods are selected properly. The results have been discussed very thoroughly and indicate a role of the RNAi pathway during host infection and present evidence for the functional relevance of the RNAi machinery on “in planta” development of the fungus Z. tritici. Moreover, the Authors did not observed any effect of RNAi on the transmission of the accessory chromosomes in Z. tritici. Discussion is scientifically mature and insightful in this matter.
Author Response
We thank the reviewer for the kind comments.
Reviewer 3 Report
The authors address the role of iRNA in the life cycle of a fungal plant pathogen. Similar approach was already undertaken in two previos studies that are cited along the manuscript. From the results obtained, the authors conclude that the tested components of iRNA machinery are not involved in fitness and evolution of the pathogen. In contrast to what was previously described, the authors claim that Ago1 is involved in virulence. Although I think that the authors show evidences for this, I think that it would be important to show that the reduced virulence of Ago1 mutant was demonstrated in the three experiments performed. If this will be the case, the reduced virulence of the mutant would be observed in each of the experiments independently.
In addition, the authors show that there are two alleles of Ago1, Ago3 and Ago4. Based on the phylogenetic tree they claimed that the genes were introgressed. I do not think that the data presented demonstrate this conclusion since there are only a few alleles of the other Zymoseptoria species and the origin of the two alleles is not demonstrated. Therefore, I think that this claim cannot be made unless the authors can show more evidences for this.
I think that there is a mistake in line 275. Was the complementation perform with Ago1 or Ago2?
Figure S6D needs to include error bars.
Figure 2: at what time was pycnidia production scored?
Round 2
